# Towards interpretable co-speech gestures synthesis using STARGATE

Louis ABEL
Université de Lorraine, CNRS, Inria, LORIA, F-54000
Nancy, France
louis.abel@loria.fr

Vincent Colotte
Université de Lorraine, CNRS, Inria, LORIA, F-54000
Nancy, France
vincent.colotte@loria.fr

Slim Ouni
Université de Lorraine, CNRS, Inria, LORIA, F-54000
Nancy, France
slim.ouni@loria.fr

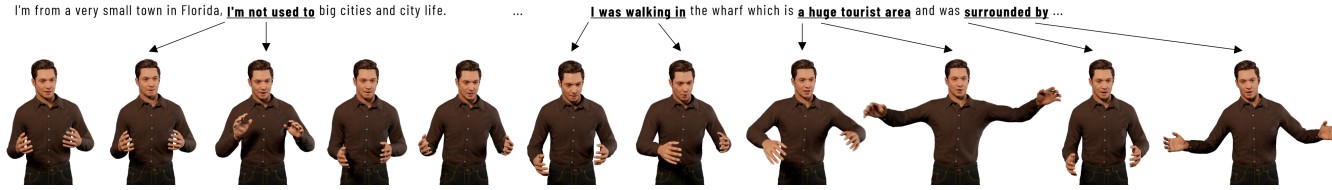

**Figure 1: An example of ECA using motion generated by our STARGATE network. This illustrates its capabilities of generating different types of gestures such as iconic gestures as depicted in this example. "I'm not used to" is illustrated with hand motion "pushing back" from the avatar. "I was walking in" is partially illustrated by a horizontal motion of both arms, although the gesture itself is "aborted" and not complete. "a huge tourist area" is illustrated by a wide opening from both arms. "Surrounded by" is illustrated by an horizontal opening from both arms.**

## ABSTRACT

Co-speech gestures synthesis is a growing field of research. However, new systems often use complex or heavy architecture, making them unsuitable for incorporation into Embodied Conversational Agents (ECAs) or for interpretation in other research fields such as linguistics, where the link between speech and gestures is difficult to investigate manually. This paper presents STARGATE, a novel architecture for Spatio-Temporal Autoregressive Graph from Audio-Text Embeddings. The model takes advantage of autoregression to provide fast generation capabilities. Additionally, it employs graph convolutions coupled with attention to incorporate explicit structural prior knowledge and enable efficient spatial and temporal processing. The model was evaluated against a state-of-the-art model in both perceptive and quantitative studies. We demonstrated that our model is capable of generating convincing gestures in the same range as state-of-the-art. Furthermore, we conducted in-depth analysis that show how our model actually produces gestures from its input.

## CCS CONCEPTS

• **Computing methodologies** → **Procedural animation**; **Natural language processing**; *Motion processing*; • **Human-centered computing** → Human computer interaction (HCI).

## KEYWORDS

Deep learning architectures, Embodied interaction, Gesture processing

**ACM Reference Format:**
Louis ABEL, Vincent Colotte, and Slim Ouni. 2024. Towards interpretable co-speech gestures synthesis using STARGATE. In *INTERNATIONAL CONFERENCE ON MULTIMODAL INTERACTION (ICMI Companion '24), November 4–8, 2024, San Jose, Costa Rica.* ACM, New York, NY, USA, 9 pages. https://doi.org/10.1145/3686215.3688819

## 1 INTRODUCTION

Co-speech gesture synthesis is an emerging field that has garnered significant attention in recent years. While the precise mechanisms underlying human gesture generation and its relationship to speech remain elusive, researchers have made considerable strides in developing techniques for generating gestures from speech data, encompassing both spoken transcripts and acoustic signals [2, 9, 21]. Despite the lack of a definite understanding of the correlation between speech and gestures, the ubiquitous presence of gestures in human communication underscores their importance in replicating natural human interactions [16, 27, 29].

To capture the essence of human gestures and incorporate them into artificial communication systems, several works were analyzing and classifying gestures. Initially, rule-based systems were used to develop Embodied Conversational Agents (ECA) [6], drawing upon insights from neuroscience and linguistics. The early systems were rudimentary and often inconsistent with findings from diverse literature sources. The lack of a unified classification scheme for gestures [5, 17, 28] and the disparate conclusions regarding the relationship between gestures and speech ([8, 19, 30]) within these frameworks hindered the development of consistent and reliable rules.

In recent years, data-driven approaches have emerged as a promising avenue for implicitly extracting the intricate patterns and rules governing the relationship between speech and gesture. These approaches use a variety of architectures, ranging from simple autoencoders [20, 31] to variational autoencoders (VAEs) and conditional VAEs [23, 25], to cover a broader distribution of gestures and a better conditioning from speech input.

In the literature, the work of Alexanderson et al. [2], StyleGestures, stands out for its innovative autoregressive architecture using normalising flow [13]. This particular network structure has gained widespread recognition as an effective benchmark for evaluating the performance of gesture synthesis systems, as evidenced by its extensive adoption in subsequent research [3, 4, 23]. Notably, it was selected as the baseline model for the GENEA Challenge [22], a challenge aimed at advancing the state of the art in gesture synthesis.

Diffusion-based systems [3, 9, 33, 34] have also garnered significant attention, producing high-quality gesture sequences but suffer from a complex architecture leading to slower processing times, giving a real trade-off between quality and speed.

In addition, while different theoretical frameworks propose diverse gesture classification, the deep learning community predominantly adopts McNeil's taxonomy [28], which encompasses the following gesture categories :

- Iconic gestures: Concrete illustrative movements that represent characteristics of elements present in the semantic content of speech.
- Metaphoric gestures: Abstract illustrative movements that metaphorically represent characteristics of the semantic content within discourse.
- Deictic gestures: Referential movements, often involving pointing to abstract or concrete elements, imparting linguistically relevant directionality through movement within discourse.
- Beat gestures: Rhythmic and undulatory movements that lack specific semantic meaning but contribute to interaction and are linked to speech.

The principal objective of gesture synthesis is to generate gestures that appear natural and human-like, encompassing all gesture categories. However, as beat gestures constitute the majority of produced gestures and are relatively simpler in form, most deep learning-based models tend to focus primarily on generating beat gestures. This emphasis on beat gestures is understandable given their prevalence in human communication. Nevertheless, the remaining part of the gesture spectrum holds immense significance in conveying meaning and enhancing the overall effectiveness of communication.

Despite the extensive research on co-speech gesture synthesis, there is a concerning lack of investigations into the underlying mechanisms that enable these methods to generate gestures, regardless of their coherence or complexity. This lack of explainability poses a significant challenge in a field that seeks new theoretical frameworks to delve deeper into the intricate relationship between speech and gesture. Deep neural networks have proven adept at extracting meaningful representations from complex data. However, in the realm of gesture synthesis, the intricate nature of gestures

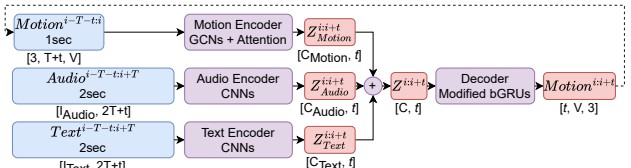

**Figure 2: An overview of the STARGATE network with its encoder-decoder structure. Our network use 3 separate encoders to process all 3 modalities separately and a unique decoder to generate motion from a multimodal latent representation. Numbers between brackets depicts tensor shapes: $T$ being the half-window length, $V$ the number of joints, $t$ the chunk size, $C_x$ the latent feature size, $I_x$ the input feature size.**

and the often opaque architectures of deep neural networks hinder our understanding of the underlying mechanisms responsible for the gesture generation. To address this challenge, one can consider exploring simpler and more interpretable mechanisms, such as graph convolutions [18]. Motivated by their successful application in the related field of locomotion synthesis, where motion is generated without the involvement of speech (e.g., walking, dancing, fighting), graph convolutions hold promise to inject a prior structural information for both enhancing our understanding of deep learning network behavior and enabling the creation of meaningful latent representations of gestures.

Inspired by these advancements, we propose a novel network architecture that aims to address the aforementioned limitations in gesture synthesis. Our contribution seeks to achieve three key objectives:

- Exploiting graph convolutions for explicit gesture structure, by integrating it into a deep neural network for gesture generation
- Efficient design using an autoregressive architecture, to accommodate speed-critical applications, such as in ECAs.
- In-depth interpretability analysis to understand how such architecture generate gestures.

In the following sections, we describe our novel architecture and the mechanisms used, followed by a comprehensive evaluation of our model against StyleGestures model using both quantitative metrics and subjective studies. While this architecture was presented in our previous work [1], this work describes it more completely and proposes a novel in-depth exploration of the behavior of our model and how gestures are produced by it, discovering hidden mechanism produced by our self-supervised model. We conclude by discussing potential future directions for our model.

## 2 METHODS

We propose a novel architecture named STARGATE (for Spatio-Temporal Auto-Regressive Graph from Audio-Text Embeddings), an overview is depicted in Figure 2. We follow an encoder-decoder structure, with a chunked-autoregressive approach. This translates to a network that takes 3 different modalities as input :

- **Audio**: A window of 1s of past and 1s of future speech.

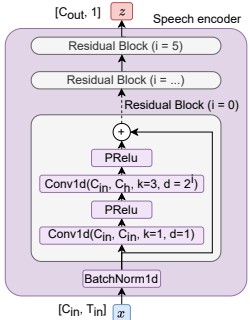

**Figure 3: The speech encoder, used for both audio and text encoding separately. Conv1D parameters have the following meaning : input channels, output channels, kernel size, dilation size.**

- **Text**: A window of 1s of past and 1s of future words.
- **Motion**: A history of 1s of past motions.

Having such a long context window is motivated by the fact that gestures are a slow modality, with an average gestures duration of 1-2s depending on if the gestures refers to a single word or a full sentence [11]. Moreover, most gestures are prepared by anticipation, which requires a broader context window than more traditional audiovisual synthesis such as lipsync or voice synthesis. Each modality as a dedicated encoder to produce a particular latent space representation, which is then fused together to create a multimodal representation of speech/gestures. This representation is finally decoded into a $t$ frames long chunk of next gesturer poses. We chose chunk output instead of frame-by-frame output to provide the network with greater flexibility in generating gestures, avoiding excessive reliance on autoregressive motion history, while also ensuring more efficient computations. The first second of motion is a sequence of zeros to start the autoregressive loop.

## 2.1 Speech encoders: audio and text

Speech can be separated in two principal components: acoustic and linguistic content. The acoustic signal produced during speech carries a lot of information including prosody (comprising energy, pitch, rhythms, etc.) and emotional state. While the linguistic content, is acoustically expressed in the speech signal, its explicit representation in textual representation enables easy exploitation.

Such representation conveys more information about the semantics content of the speech than raw acoustic informations, text a crucial source of information to model iconic, deictic and metaphoric gestures which are all directly linked to this semantics content, while the latest gesture category, beat gestures, are linked to the acoustic signal. Both modalities are thus needed to generate both dynamic and meaningful gestures. In our architecture we use those two modalities through two similar but separate CNN-based encoders, its implementation is depicted in Figure 3. The input feature is first passed through a batch normalization layer before going through one of the seven residual blocks. Each block consists of one classic convolution and a dilated convolution, progressively compressing the temporal dimension of the input to a single vector representation. While encoders follow the same architecture, audio

input and text input are not the same size, thus only convolution channels are different between the audio and text encoder.

*2.1.1 Audio encoder.* For the audio modality, we extracted 27 MFCC coefficients at a temporal resolution of 120Hz to match dataset temporal resolution, but are downsampled at 60Hz as such high resolution is not needed to generate a slow modality such as gestures. We then fed the encoder a 2s sliding window, with 1s of past information and 1s of future information. Beat gestures are heavily linked to the acoustic signal thus it is the role of the audio encoder to extract a latent vector that enables the decoder to understand audio-controlled parameters such as speed and range of motions. The channels used for each convolution in the audio encoder are the following, first one being $C_{in}$ in Figure 3, latters being $C_h$ : 27 (input), 64, 96, 128, 128, 256, 256.

*2.1.2 Text encoder.* For the text, we choose to replace each word of the transcription by its BERT embeddings [10], as it has been shown that they are a powerful and compact way to represent text content. However such embeddings represent only words with their context, but without any rhythm information, which expose the problem of features synchronization. For the model to effectively uses both audio and text modality together we needed to align BERT embeddings with audio features extracted at 60Hz. To do that we duplicated those embeddings with respect to each words timing using forced alignment, in our case with Montreal Forced Aligner [26]. This allows the network to have both meaningful semantic features from the BERT embeddings, but also the pace of the text itself. The same 2s context window, with 1s of past embeddings and 1s of future embeddings is then fed to the encoder. This explicit semantic information is crucial for generating complex gestures such as metaphoric, iconic or deitic gestures that directly refer to the linguistics content of the speech. As they can depict particular concept or shapes, a deep understanding of language, gestures and the link between both is thus needed by the network to create such gestures. The channels used for each convolution in the text encoder are the following : 768 (input), 768, 768, 512, 512, 396, 396.

## 2.2 Motion encoder

One of the benefits of using an autoregressive approach is that motion, being the output, can also be part of the input. Having motion as the third modality helps to keep a good consistency for the gestures trajectories but also to create a speech-gestures multimodal representation. Our motion encoder is based on the work of [35], an overview of the motion encoder can be seen in Figure 4. The input motion is represented as exponential maps [12] as they have the advantage of being a continuous representation of rotation in comparison of Euler angles, and are more compact than quaternions. It goes through a batch normalization layer and then through 3 ST-GCN (which stands for Spatio-Temporal Graph Convolution Network [35]) chained together to process motion data using graph convolution network. Such structure allows producing strong embeddings capturing both the spatial and the temporal links inherent to motion data. To our knowledge this is the first work in the field of co-speech gestures synthesis using graph convolutions to inject a more explicit representation of gesture.

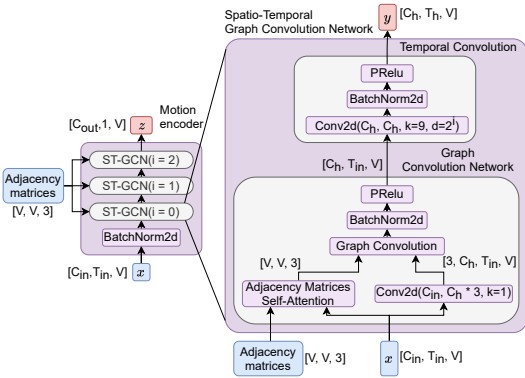

Figure 4: The motion encoder consists of 3 chained ST-GCN blocks, each of them using the same adjacency matrices. V is the number of nodes in the graph, see Figure 5. Next to it is an ST-GCN block, which chains a spatial transformation through a self-attention mechanism before a graph convolution and a temporal transformation. See Figure 3 for Conv2D parameters meaning.

*2.2.1 Graph Neural Network.* In order to both generate convincing gestures and provide insight into the network's process for producing gesture latent representation, we used multiple mechanisms within our motion encoder. The first mechanism involves the use of Graph Convolution Networks (GCNs) [18] instead of classic CNNs. This clearly contrast with classic approach of using CNNs without constraints on the spatial organization of information. Our aim is to force the network to understand the anatomical and physical constraints of the human skeleton, potentially leading to a more explicit and comprehensible gesture latent space.

GCNs represent a specific type of convolutional neural network that treats data not as a grid layout in N dimensions, but as an undirected graph. This approach enables the incorporation of explicit neighboring information to emphasize the significance of adjacency within a graph. Unlike classic convolutions, where the network must discern and establish relevant connections between data, GCNs possess prior knowledge of these connections through the graph structure. Consequently, only relevant nodes are aggregated by the convolutional process directly from the outset of training. In practice, this translates to the use of the graph adjacency matrix to determine which nodes to use for each computation, rendering GCNs as efficient as CNNs.

In our case we used the ST-GCN block from [35], depicted in the Figure 4, it performs both a spatial transformation using a graph convolution, and a temporal processing using a Temporal Convolution Network (TCN). This network makes the use of multiple adjacency matrices, each of them with a particular set of links between nodes to model a special relationship :

- **Self-link**: this matrix allows information to stay in the same node during computation.
- **Neighborhood**: this models the direct neighborhood between nodes, the further away the neighbor is from a node, the less impact it has on it during the computation, so we want only the direct neighbors to contributes to the next state of a node.

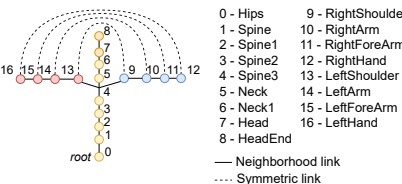

Figure 5: The graph we used, with 17 nodes, neighborhood links (direct links) are in solid lines while symmetric links are in dashed lines. Self-link are omitted for a simpler representation.

- **Symmetric**: this last matrix is for symmetrical neighborhood, this symmetrical adjacency allows direct flow of information between the left and right arm. Note that symmetric links does not forces the network to output symmetric gestures, its simply allows information to goes from left to right part of the body more directly.

We illustrate the graph used in our network in the Figure 5, with 17 nodes and links between the nodes. Motivated by the fact that we want to create new gesture representation, those matrices are initialized at the start of the training, as a prior knowledge, but are left unfrozen and as parameters for the network to make changes in matrices according to the task of gesture synthesis, potentially creating interesting new links between nodes.

The second advantages of using graph is its robustness to potential changes in the skeleton, both in terms of physical changes (e.g., multi-speaker training with different body) but also in layout changes (e.g., changing dataset). The graph convolution will make the extracted features a lot more invariant to those changes as a human skeleton will always keep a similar layout (shoulder linked to an arm, then forearm, ...).

*2.2.2 Attention mechanism.* In the ST-GCN implementation of [35] and ours, there is a self-attention mechanism on adjacency matrices before the graph convolution. The input motion data is passed through a scaled dot-product self-attention block, to create an 'attention matrix', one for each adjacency matrix. Those 'attention matrices' are added on top of the base adjacency matrices to produce what we call 'dynamic adjacency matrices'. This is motivated by the fact that even if we let the network make small changes to adjacency matrices during the training, at inference time it would remain static. This attention mechanism allows introducing dynamic modification at inference, to make the network pay more attention to particular parts of the body for each chunk of generated frames.

## 2.3 Motion decoder

Audio, text and gesture latent space are then combined, in our case simply by concatenation, to produce a *t*-frames multimodal latent space which is fed to the motion decoder. The decoder, which consists of stacked RNNs (in our case Gated Recurrent Units, GRUs, networks [7]), will output the next chunk of *t*-frames skeleton pose, those frames are then used to compute the next batch of frames. This decoder is depicted in Figure 6 The major drawback of autoregression is to work only with previous information, and not being

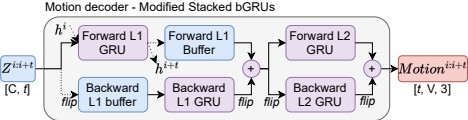

**Figure 6: Our decoder uses a stacked GRUs approach. We implemented bidirectionality using buffers at first layer (L1) GRUs, the forward one storing forward L1 hidden features, the backward one storing longer latent sequence. Only forward L1 GRU reuse its hidden states during computation as others do not have continuity in their inputs.**

able to analyze a full sequence of gestures. As a consequence we could not use bidirectional GRUs to get an in-depth comprehension of the whole gestures sequences. But motivated by the benefits that it could bring and as we are generating batch of frames, we can use bGRUs on those partial sequence. In fact, we can also extend the processed sequence by storing previous generations inside a forward and a backward buffer, in our case both buffer store 128 frames of previous information. As continuity cannot be kept on later layers, we only keep track on the first forward GRU layer hidden state. This bidirectional implementation should allow the network to learn relationship between past and future information present in the multimodal representation.

## 3 TRAINING

### 3.1 Dataset : BEAT

We trained all our models on the BEAT [24] gesture dataset. It provides both high quality and large volume of multimodal data, ranging from audio, word and phoneme-level transcriptions, motion capture data for body, hands, and facial expressions. We used the data corresponding to the speaker 1, giving us 4 hours of data, which we split into training, validation and test sets using a 90/5/5 ratio. Audio and motion data preprocessing follows the protocol and code proposed by StyleGestures [2]. We also augmented data by using the same mirroring strategy, which allow us to double the amount of data.

### 3.2 Loss

Our model is trained to minimize two terms : one Huber [15] loss on exponential map and one Huber loss on positions, which are computed from the exponential maps. This is motivated by the fact that only minimizing the exponential map error as the network objective will give an equal importance to each joints in the skeleton, however, as the skeleton is inherently a hierarchy, we want a near-perfect control of the hips and spine, as they will have an impact on all end effectors. The loss is thus defined as follows:

$$Loss = \mathcal{H}(r, \hat{r}) + \mathcal{H}(p, \hat{p})$$

With $p$ and $r$ respectively, the positions and exponential map of the reference sample, $\hat{p}$ and $\hat{r}$ respectively the positions and exponential map of the generated sample and $\mathcal{H}$ the Huber loss.

## 4 EVALUATION

### 4.1 Quantitative metrics

This section presents evaluations of our proposed model and a variant, "Audio Only," which excludes the text encoder. Those two models are compared against state-of-the-art model. We compare the performance of these models against a well-established benchmark in the field, StyleGestures [2]. This choice is motivated by StyleGestures' autoregressive architecture and its frequent adoption as a reference model for gesture synthesis research [22], while it is not the current SOTA model compared to more recent diffusion-based approach, it allows fair comparison between 2 similar architectures. We show enhancements on a well known autoregressive approach which have the advantage to be more tractable than diffusion, known to be slow both at training and inference.

**Frechet Gestures Distance (FGD).** Evaluation in the field of gesture synthesis is heavily reliant on subjective assessments. As we are dealing with generative models and gesture is a complex modality where a single gesture can be used in multiple contexts and a single speech segment can be accompanied by one or more gestures, traditional metrics (such as Mean Angle Error or Root Mean Square Error) designed for one-to-one or one-to-many mappings cannot be applied effectively in this many-to-many setting.

The best attempt to get an objective metric for gesture synthesis is inspired by the Frechet Inception Distance (FID) in image synthesis [14]. The FID is calculated by measuring the Frechet distance between latent features extracted from a set of ground-truth samples and generated ones produced by an Inception network. This score has been adapted for the task of gesture generation in [32], now known as the Frechet Gestures Distance (FGD). To compute the FGD, we needed to retrain the Inception network proposed by [32] with our dataset as it differed significantly from what the available network has been trained on. As a result, our FGD results cannot be directly compared to those of other papers that report FGD values for StyleGestures. The advantage of this method is that the Inception network behaves as an unbiased evaluator, resulting in a metric that is more closely aligned with actual human perception.

As we can see in Table 1, both STARGATE variants outperforms StyleGestures. Interestingly, the Audio Only variant of our system exhibit lower FGD values compared to our baseline model. This behavior could be attributed to the fact that our baseline model is the only variant capable of generating convincing non-beat gestures (such as the ones depicted in Figure 1, see Section 5.1), albeit in small numbers. These gestures are significantly more challenging to master and often deviate significantly from the reference gestures, contributing to higher FGD scores despite not providing superior perceptual quality due to their limited occurrence.

**Performance.** While performance should not be a primary concern when designing a model for gesture synthesis, we aimed to develop a network capable of running in scenarios where speed is critical(e.g. for ECAs), generating convincing gestures as quickly as possible. To assess performance in this context, we conducted benchmarks that consider preprocessing steps, as they can significantly impact computational overhead (such as BERT embedding computations). Therefore, all reported timings are based on raw waveform/sentence input, with a batch size of 1. We conducted

| | Nb params | Graph? | Audio? | Text? | FGD ↓ | Inference time ↓ [Time per frame ↓] | | | |
|---|---|---|---|---|---|---|---|---|---|
| | | | | | | 5s | 10s | 30s | 80s |
| StyleGestures | 82M | ✗ | ✓ | ✗ | 14.15 | 7.76s [90ms] | 12.90s [70ms] | 31.05s [50ms] | 80.07s [50ms] |
| STARGATE | 43.5M | ✓ | ✓ | ✓ | 10.58 | 6.51s [37ms] | 8.31s [17ms] | 13.40s [8ms] | 23.78s [5ms] |
| STARGATE *Audio Only* | 30.9M | ✓ | ✓ | ✗ | **8.61** | **3.49s [19ms]** | **3.98s [8ms]** | **6.13s [3ms]** | **10.68s [2ms]** |

**Table 1: Results of quantitative comparison using FGD and models benchmarks according to the duration of the utterance (5s to 80s). Note that StyleGestures outputs 20fps while our model outputs 60fps. Bold values depict best model. Benchmark used the following hardware configuration: i7-11850H and NVIDIA RTX A3000 Laptop.**

benchmarks for different sample lengths as some neural network architectures exhibit performance advantages over longer sequences. For each sample length, we performed multiple runs and report the average. To account for the fact that both systems generate gestures of different lengths (depending on framerate, computation windows, etc.), we also report execution time per frame to facilitate a fairer comparison.

As for the FGD, both STARGATE variants provide the output consistently faster than StyleGestures. In short 5s sequence generation, our model performs up to 1.4x faster than the input length, while in 80s long sequence generation, our model performs up to 7.5x faster than the input length. In comparison, StyleGestures is 1.5x slower in the first scenario and remains neither faster nor slower in the second scenario. However, this model outputs 20fps gesture sequences, whereas ours output 3 times more frames at 60fps. Thus, when considering the time per frame, our model is 4.7x faster than StyleGestures per frame generated in the short sequence scenario, and in long sequence generation, our model takes advantage of parallel processing of input modalities (as audio and text are not in the autoregressive loop), becoming 25x faster per frame generated for our Audio Only variant, and 10x faster for our standard model (audio + text).

In both audio-only and text-integrated cases, this implies that integration into ECAs can be achieved, allowing for more natural interaction with future human machine interaction with gestures-enabled avatars in low latency scenarios.

## 4.2 Subjective Evaluation

To further evaluate our model, we conducted a Mean Opinion Score (MOS) evaluation. to assess the overall quality of gestures generated by our novel architecture. We rendered 3D animations using the GENEA Challenge model [22] to align with their evaluation protocol as closely as possible. All stimuli were presented at a frame rate of 60 fps. The study were conducted using Prolific, a crowdsourcing platform that facilitates easy recruitment of participants. The study incorporated control samples to identify and eliminate participants who failed attention checks. All participants were selected from the United States, had English as their native language, and were compensated for their participation in our studies.

Inspired by the GENEA Challenge evaluation protocol, we evaluated both the "human-likeness" and "appropriateness" of the generated gestures, using slightly modified questions to gain a clearer understanding of the strengths and weaknesses of our model's gesture generation capabilities.

The evaluation was divided into two parts. The first part involved viewing videos without audio and answering the question,

| Model | Human-like ↑ | Credibility ↑ | Consistency ↑ |
|---|---|---|---|
| Reference | 6.19 ± 0.28 | 5.27 ± 0.23 | 5.16 ± 0.23 |
| Mismatch | N/A | 4.92 ± 0.20 | 4.77 ± 0.22 |
| StyleGestures | 5.97 ± 0.25 | 4.87 ± 0.22 | 4.70 ± 0.23 |
| STARGATE | 5.89 ± 0.28 | 5.0 ± 0.20 | 4.85 ± 0.22 |

**Table 2: Results of our MOS evaluation, we report the mean and a 95% confidence interval for each aspect.**

"How human-like does the gesture motion appear?" The second part involved watching videos with audio and responding to two questions: "How credible are the gestures with respect to the speech?" and "How consistent are the gestures with respect to the speech?" Participants were asked to rate each question on a scale of 1 to 7. We evaluated four gesture generation systems in this study: Reference (ground truth), Mismatch, StyleGestures, and STARGATE.

Mismatch was created by using synthetic motions from the STARGATE network and pairing them with audio from a different sample. As a result, it was not included in the first part of the evaluation, where no audio was present. We presented 30 videos for each system, each lasting 9 seconds and we had a total of 25 participants (12 female and 13 male).

The results are shown in Table 2. As can be seen, StyleGestures achieved slightly higher score than our STARGATE model in the overall human-likeness aspect, but our model is slightly better in both consistency and credibility when audio is available. We attribute this difference without audio to the presence of non-beat gestures in our model, which are not always produced clearly (as evident in the aborted gestures in Figure 1). This inconsistency sometimes results in a mix of iconic and beat gestures, leading to a perception of unnaturalness when the gestures are not accompanied by audio cues.

Table 2 also corroborate findings from previous research [22], where Mismatch exhibits higher ratings compared to StyleGestures and our model. We attribute this observation to the high prevalence of beat gestures in the dataset and the generated gestures. Beat gestures are inherently consistent and credible when they align with the audio rhythm. This is true for both mismatch and non-mismatch motions, as they both originate from the same model output, which is capable of matching the overall pace of the dataset. Consequently, the gestures in both scenarios were able to convince users. In contrast, the reference motion exhibits more consistent and credible gestures due to the presence of highly semantic gestures presented to the users. This highlight the problematic of evaluating complex gestures generation by novice who can be fooled by simple beat gestures.

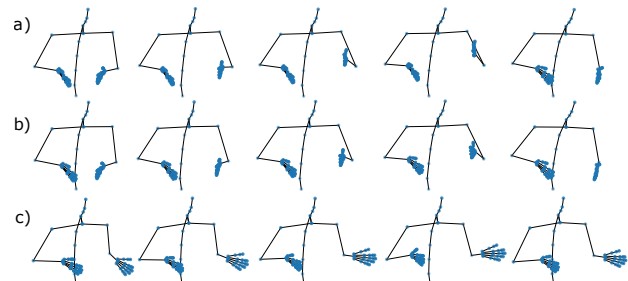

**Figure 7: This illustration showcases three iterations of the sentence "To take down an evil king" generated by different variants of the STARGATE model. Each gesture lasts 1 second. a) Utilizes the standard STARGATE architecture and generates an iconic gesture representing "take down" with precise timing. b) Utilizes a text-only variant and produces identical gestures. c) Utilizes an audio-only variant and generates a beat gesture, indicating a lack of semantic understanding by the network in this instance.**

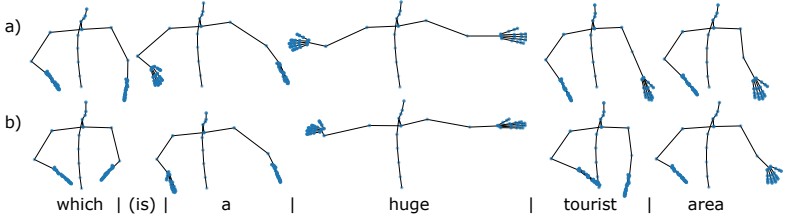

**Figure 8: a) illustrates the generation of the sentence "Which is a huge tourist area" using STARGATE. b) depicts the generation of each individual word, with its BERT embeddings stretched over 2 seconds, as it is the minimum input length for STARGATE. Nearly the same poses are generated, demonstrating how STARGATE makes the relationship between a word and a pose.**

## 5 INTERPRETABILITY

As we collaborate with researchers in more theoretical fields such as linguistics and cognitive science, one of our objectives was to develop a model whose behavior we could analyze thoroughly, understanding its internal workings to advance these fields and introduce new theoretical frameworks to our community. This led us to create a deterministic model instead of a more advanced probabilistic one. While the latter could produce a wider range of gesture outputs, its interpretability would be significantly more complex.

In this section, we will present the results and findings obtained from conducting a set of exploratory tests on our STARGATE model. To our knowledge, this is the first time that a co-speech gesture synthesis model has been analyzed in such a manner.

### 5.1 Importance of text and rhythm

Our first experiment aimed to understand the type of information extracted from each input modality. To achieve this, we generated our test set with multiple variants of our network. We tested networks trained with all modalities but some modalities set to zero, and networks trained with only one modality. Our observations revealed that regardless of the configuration, our STARGATE network predominantly utilizes the text input in conjunction with the rhythm from the forced alignment step. Interestingly, the rhythm appears to be the sole information extracted from the audio input. We attempted replacing MFCC with alternative audio features like F0 and energy, yet achieved similar outcomes. Thus, our model relies solely on text and rhythm to generate both beat and iconic gestures. We can see in Figure 7 that only variants utilizing the text

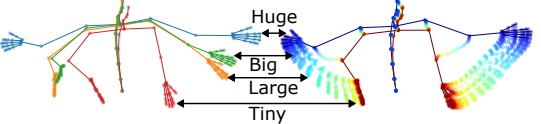

**Figure 9: On the right is the stroke pose of the sentence "Which is a huge tourist area" in blue and "Which is a tiny tourist area" in red, in between colors are the generation of the interpolation between BERT embeddings. On the left we found the same in-between gestures with concepts such as "large" and "big" instead of huge.**

are able to produce iconic gestures, while the audio-only leads to pure beat gestures.

### 5.2 Word to pose

We then investigate why text and rhythm sufficed for the model. We observed that altering the rhythm by stretching or compressing certain BERT embeddings did not result in quicker or slower gestures. Since the training data established a precise pace for gestures, the network learned this dynamic. Consequently, artificially changing the word rhythm led to gestures maintaining a particular pose if the stretch was too prolonged or no gestures at all if the compression was too severe.

However, when we generated gestures using only one word (and thus one BERT embedding, still computed with the entire sentence context) stretched over a 2-second sequence (to match our minimal input length), we discovered that our model assigned a particular pose for each BERT embedding vector. In essence, our model generates gestures by creating a pose for each different

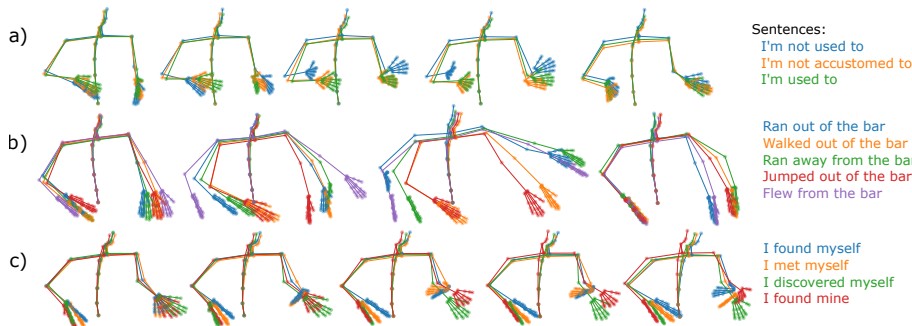

**Figure 10: Examples of generation with slightly varied sentences to illustrate that similar concepts yield similar iconic gestures. a) This demonstrates that our network struggles with negation, as blue and green depict the same gesture despite conveying inverted concepts. b) This illustrates that distinct concepts such as jumping and walking result in dissimilar gestures, while running and fleeing elicit the same gesture. c) In this example, the same hand gesture is depicted in blue and orange, but no gesture is produced in green, as the word "discovered" alters the context significantly. Additionally, in the case of red, the gesture is delayed and synchronized with "mine," whereas others are tied to the verb.**

BERT embedding and links them through interpolation based on the word rhythm. We illustrated this phenomenon in Figure 8, where we display both a gesture generated from a sequence of BERT embeddings for the entire sentence and multiple generations, each with only one BERT embedding for one word.

## 5.3 From word interpolation to gesture interpolation

As our model assigns a specific pose to each BERT embedding, we can interpolate between two BERT embeddings, indirectly interpolating in the final gesture space. In Figure 9, we use the example sentence "Which is a huge tourist area" from our test set. This sentence triggers an iconic gesture linked to the word "huge"; however, replacing "huge" with "tiny" results in our model not generating such an iconic gesture. By interpolating between the embeddings of "huge" and "tiny," we generate a continuous spectrum of gestures ranging from a wide opening to no gesture at all.

Further exploring this behavior, we discovered that this interpolation unveils other semantic concepts. When we substituted synonyms of "huge," such as "large" or "big," we obtained similar gestures positioned perfectly in the middle of the interpolation. Despite carrying the same meaning, these words have a lesser impact than "huge," resulting in a narrower arm opening.

It is noteworthy that we did not observe a specific iconic gesture for "tiny" (compared to a beat gesture). This might be explained by the fact that the concept of "tiny" or a similar iconic gesture had not been encountered during training, even though the BERT embedding adequately encoded the "tiny" concept within its context.

## 5.4 From similar words to similar gestures

This revealed a clustering effect between the BERT embeddings and their associated poses, wherein similar words are interpreted as the same concept by our network, leading to the generation of a consistent group of poses. To validate this observation, we generated variants of sentences from our test set sequence containing iconic gestures. Different samples are illustrated in Figure 10.

We observed that altering the sentence while preserving the same meaning resulted in the reproduction of identical gestures. Just as the BERT encoder generates close embeddings for similar words in the same context, our network replicates this effect with gestures, portraying synonyms and semantic concepts through similar gestures.

However, due to the limitations of our training data, numerous concepts lack associated iconic gestures (refer to Figure 10b). Similarly, certain linguistic transformations, such as negation, result in the production of identical gestures (as seen in Figure 10a), despite significant differences in semantics.

## 6 CONCLUSION

We introduce STARGATE, a novel chunked autoregressive architecture that uses three input modalities to construct a unified latent representation of speech-gestures and subsequently generate gestures. This architecture represents a significant advancement by using graph convolutions instead of traditional convolutions, thereby explicitly integrating prior knowledge of the human skeleton structure. By integrating anatomical and structural constraints, it ensures the avoidance of potential unnatural gestures. and potentially enhancing interpretability.

Our evaluation, encompassing both quantitative metrics and subjective studies, demonstrates the model's capability in generating diverse gestures, spanning from basic beat gestures to more intricate ones like iconic and metaphorical gestures. This may suggest that the model effectively combines linguistic information with learned insights into the coordination of speech and gestures. The comprehension and interpretability of our network pave the way for advanced training methodologies that leverage the word-to-pose effect. Furthermore, this enhances our understanding of the mechanisms underlying the generation of co-speech gestures in human speech. These pioneering results will undergo further in-depth analyses, particularly utilizing a larger and more diverse corpus of co-verbal gestures, to ensure robust validation of the findings.

We are now working on extending our model to include both finger gestures and multi-speaker generalization.

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
