# OpenReview forum: "Towards interpretable co-speech gestures synthesis using STARGATE"
_ACM.org/ICMI/2024/Workshop/GENEA — GENEA Workshop 2024_

### Official Review · Reviewer_VSo5 · 2024-07-17
**Applying spacio-temporal graph convolution to co-speech gesture generation is novel**

**Rating:** 9
**Confidence:** 5

**Review:**

#### SUMMARY
This paper aims to develp co-speech gesture generation method using graph neural network.
Applying spacio-temporal graph convolution to co-speech gesture generation is novel, and it is reasonable to have constraint of human body structure to generate human movements.
The model uses separate encoders for audio, text, and motion data, and combines these to create a mutimodal latent representaion. The motion encoder incorporates spatio-temporal graph convolution network.
Quantitative metrics showed the proposed method outperformed StyleGesture (normalized flow based method).
Subjective studies showed the proposed method can generate gestures as natural, credible and consistent as StyleGesture.

#### STRENGTHS
- Applying graph neural network to co-speech gesture generation is novel
- Both Quantitative and Subjective studies were conducted
- The method has high reconstruction ability and is faster than normalized flow based model (probably faster than diffusion model).
- The paper is well-organized, the technical descriptions are well-written, and the experiments are reproducible.

#### WEAKNESSES
- The paper should compare the proposed method with some diffusion base state-of-art methods.

**Nominate For A Reproducibility Award:**

graph neural network approach should be explored!

---

### Official Review · Reviewer_utkg · 2024-07-26
**Exploration of using GCN is interesting**

**Rating:** 8
**Confidence:** 4

**Review:**

This paper proposes an encoder-decoder structure using a Graph Convolutional Network (GCN) for a co-gesture synthesis system, incorporating physical constraints for each joint. The audio and text encoders are conventional CNNs, while the motion encoder is a Spatial-Temporal Graph Convolutional Network (ST-GCN) with an attention mechanism. The motion decoder is a modified bi-directional GRU, enabling autoregressive motion generation. The system processes a window of audio, text, and previous motion inputs in the encoder, outputting the next motion segment. Both subjective and objective experiments demonstrate that this approach offers comparable performance and faster inference speeds compared to StyleGesture.

Pros:
1. The paper explores using ST-GCN with physical constraints by providing joint connection adjacency matrices.
2. The results show that their CNN-based approach can generate motion more quickly while maintaining comparable performance to StyleGesture in both quantitative and qualitative studies.
3. They demonstrate reasonable interpretability of the proposed system, which can express varying degrees of words (e.g., tiny, big, huge).
4. The paper is easy to follow.

Cons:
1. The effectiveness of applying ST-GCN remains unclear, as the evaluation results are missing.
2. The effectiveness of the text modality has been demonstrated in many previous co-gesture synthesis papers. While this aspect is satisfactory, it is less compelling compared to the novel ST-GCN.

Comments:
Personally, ST-GCN was proposed to denoise the noisy hand-pose with learnable static and dynamic weight. These weights enable the model to pay less attention to noisy and less trustworthy neighboring joints. However, it is still unclear to me which unreliable neighboring joints in the previous motion require ST-GCN for denoising. It would be more interesting to see how ST-GCN further improves performance compared to using a temporal convolutional network as the motion encoder. Additionally, although the authors claim that the designed system is supposed to be flexible with different skeletons, the evaluation on multiple and varied skeletons seems to be missing.

**Nominate For A Reproducibility Award:**

The paper descriptions are clear. If code and pre-trained models are released, it should be easy for the reader to reproduce the results.

---

### Official Review · Reviewer_g16k · 2024-07-30

**Rating:** 7
**Confidence:** 5

**Review:**

In this paper, the authors present a deep learning architecture named STARGATE for co-speech gesture generation. The model is autoregressive and utilizes graph convolutions. Together, these choices enable fast and efficient generations, as well as possibilities for interpretability and enforcing prior structure (i.e., skeletal structure). Audio, text, and motion are encoded separately and concatenated. Stacked GRUs are used for decoding.

Performance is assessed primarily against StyleGestures, both objectively and subjectively. Results are good, but there is clear room for improvement. A deeper ablation study would be nice to see in future work, to show which aspects of the architecture are most important. For example, I wonder if the inclusion of symmetric links in the graph was actually useful in the outcome or not.

The graph is explicitly structured in the shape of an upper body skeleton. Does this mean the morphology of the target character must be known and fixed in advance? How might the gestures be adapted to other skeletal structures with more or fewer joints? Would the model need to be retrained from scratch?

The interpretability analysis was most impressive. The ability to interpolate in the language space and see the gesture interpolate in the motion space was really cool. I believe this paper will make a fabulous addition to the GENEA workshop.

**Nominate For A Reproducibility Award:**

n/a

---

### Decision · Program_Chairs · 2024-07-30

**Decision:**

Accept

**Comment:**

The paper presents a novel co-speech gesture generation method utilizing a Spatial-Temporal Graph Convolutional Network (ST-GCN) within an encoder-decoder framework. The proposed system integrates separate encoders for audio, text, and motion data, combining them to create a multimodal latent representation. Quantitative and subjective experiments demonstrate that this approach offers comparable performance and faster inference speeds compared to StyleGesture.

Overall, the article presents a significant advance in the generation of co-speech gestures through the innovative use of ST-GCN, offering a faster and comparatively efficient model to StyleGesture. Although some points require further evaluation and comparison, the article's strengths in terms of analysis and novelty merit acceptance, as both reviewers point out. The article may be a candidate for the reproducibility prize for its clear descriptions and innovative approach as long as the authors supply the code.